# Current Prevalence of Oral *Helicobacter pylori* among Japanese Adults Determined Using a Nested Polymerase Chain Reaction Assay

**DOI:** 10.3390/pathogens10010010

**Published:** 2020-12-24

**Authors:** Ryoko Nagata, Tatsuya Ohsumi, Shoji Takenaka, Yuichiro Noiri

**Affiliations:** Division of Cariology, Operative Dentistry and Endodontics, Faculty of Dentistry & Graduate School of Medical and Dental Sciences, Niigata University, Niigata 951-8514, Japan; lemmings@dent.niigata-u.ac.jp (R.N.); osumi@dent.niigata-u.ac.jp (T.O.); noiri@dent.niigata-u.ac.jp (Y.N.)

**Keywords:** *Helicobacter pylori*, oral cavity, nested polymerase chain reaction (PCR), prevalence, characteristic distribution

## Abstract

In Japan, gastric *Helicobacter pylori* infection prevalence has markedly decreased with socioeconomic development. We aimed to investigate the prevalence of oral *H. pylori* in Japanese adults in 2020 by sex, age, sampling site, and medical history. Unstimulated saliva, supragingival biofilm, and tongue coating were obtained from 88 subjects–with no complaints of upper digestive symptoms–attending a dentist’s office for dental check-up or disorders. Supragingival biofilm was collected from the upper incisors, lower incisors, upper right molars and lower left molars to analyze the characteristic distribution. Oral *H. pylori* was detected using nested polymerase chain reaction. Oral *H. pylori* prevalence did not statistically differ by sex or age. Supragingival biofilm (30.7%) was the most common oral *H. pylori* niche; it was also detected in 4.5% of saliva and 2.3% of tongue samples. The lower incisor was the most common site among the supragingival biofilm samples, followed by the upper incisors, lower left molars, and upper right molars. Oral *H. pylori* DNA was frequently detected in patients with a history of gastric *H. pylori* infection. Oral *H. pylori* has a characteristic distribution independent of sex and age, suggesting that it is part of the normal microflora in the adult oral cavity.

## 1. Introduction

*Helicobacter pylori* is a clinically important pathogen that causes chronic gastritis and peptic ulcers, which are associated with gastric carcinoma [1,2,3,4]. The prevalence of *H. pylori* infection varies among countries, ranging from 19% to 88%, due to differing socioeconomic factors, such as the hygiene status and the level of industrialization of the society [5]. The principal transmission route of *H. pylori* has not been clearly defined, but there is evidence supporting gastro–oral, oral–oral, and entero–oral routes [6,7]. The oral cavity is the second most colonized site after the stomach [8].

*H. pylori* is found in the oral cavity [8], commonly existing in the saliva [9,10,11], supragingival plaque [8,9,10], dentine caries [8], subgingival plaque in chronic periodontitis [8,9,10], infected root canal [8], and the coating of the tongue [9]. Oral *H. pylori* appears to be viable but cannot be cultivated [12]. As the urease test is not specific for the detection of *H. pylori* in the dental biofilm, oral *H. pylori* detection has been mainly performed by polymerase chain reaction (PCR) analysis [8,9,10]. However, the prevalence of *H. pylori* in the oral cavity ranges from 0 to 100% [9,10]. This wide variation may be due to the characteristics of the sample population, differing sampling procedures, and the specificity and the sensitivity of the primers used [9,10]. In particular, standard PCR frequently yields false-positive and false-negative results because of its detection limits [11]. The nested PCR method is an alternative method to enhance the sensitivity and specificity of PCR [13,14,15].

The prevalence of *H. pylori* infection among Japanese children and adolescents has been decreasing in accordance with changes in standards of living over the past three decades [16]. A meta-regression analysis demonstrated a clear birth-cohort pattern of *H. pylori* infection, revealing that the predicted prevalence was 59.1% for individuals born in 1950, but only 15.6% and 6.6% for those born in 1990 and 2000, respectively [17]. Thus, the incidence of gastric cancer is expected to continually decrease, reaching a low level in the future [16].

The association between gastric and oral *H. pylori* infection remains controversial. The evidence for an association is based on epidemiological studies demonstrating a correlation between the presence of oral and gastric *H. pylori* [8]. The evidence against an association comes from the independent distribution of oral *H. pylori* in relation to the stomach infection status [13,17]. If there is a strong association between oral and gastric *H. pylori* infection, a decline in the prevalence of gastric *H. pylori* infection in the young population should correlate with a declining prevalence of oral *H. pylori*. However, there are no data on the prevalence of oral *H. pylori* in Japanese individuals by age.

The aim of this study was to investigate the prevalence of oral *H. pylori* by age (birth year) in Japanese individuals using a nested PCR assay. We also analyzed relative frequencies by gender, sampling site, and medical history.

## 2. Results

### 2.1. Relative Frequencies by Sex, Age and Medical History

Oral *H. pylori* DNA was detected in 2 of the 88 participants (2.3%) by single-step PCR, whereas nested PCR revealed oral *H. pylori* DNA in 32 of the 88 participants (36.4%) (Figure 1). There was no significant difference in the prevalence of oral *H. pylori* between men and women, despite the high proportions of gastric infection carriers and women who had received eradication therapy (*p* > 0.05, Table 1 and Table 2). The prevalence of oral *H. pylori* was not associated with age among the young, middle-aged, or elderly populations (*p* > 0.05, Table 2). The prevalence of oral *H. pylori* was significantly higher in individuals who had received eradication therapy for gastric *H. pylori* compared with those who had not (*p* < 0.05, OR = 7.95, Table 3). 

All participants had no dental caries and a probing depth (PD) of >4 mm at the sampling site for supragingival dental biofilm. The average plaque control record, according to O’Leary’s method, was 29.31 ± 17.68% for individuals who had oral *H. pylori* DNA and 25.18 ± 16.29% for negative individuals, respectively [18]. There was no significant difference between positive and negative individuals (*p* = 0.932). Mean plaque score, according to Turesky modification of the Quigley-Hein Plaque index (PI) [19], was 1.97 for positive individuals and 2.07 for negative individuals, respectively. There was no significant difference between positive and negative individuals (*p* = 0.645).

### 2.2. Relative Frequencies by Sampling Site and after Eradication Therapy

The frequencies of oral *H. pylori* detected in samples taken from seven sites in three different niches, including the saliva, tooth, and tongue, are summarized in Table 4. Oral *H. pylori* was identified in 43 of the 88 participants. The bacteria could be detected in a maximum of four individual sites, namely the upper incisors, lower incisors, upper right molars, and lower left molars. The supragingival biofilm was the most common site of detection among the three niches; *H. pylori* was detected in the supragingival biofilm of 30.7% (27/88), saliva of 4.5% (4/88), and tongue of 2.3% (2/88) of the participants (OR: 19.0, 2.0, and 1 (reference), respectively). The lower incisor was the predominant site of detection in the supragingival biofilms, followed by the upper incisors, lower left molars, and upper right molars (*p* < 0.05). 

One participant who had undergone successful eradication therapy provided a supragingival biofilm sample every two months, and the existence of oral *H. pylori* was examined using nested PCR. The results showed that oral *H. pylori* DNA was detected three out of six times, from either the upper or lower incisor sample (Appendix A). 

### 2.3. Accuracy of the Nested PCR Method

To verify the accuracy of the nested PCR method used in this study, 228-bp amplified fragments from the 43 positive samples were subjected to DNA sequencing. All sequences showed 100% homology with that of *H. pylori* IID3023 (Appendix A).

### 2.4. Detection Limits for Single-Step and Nested PCR

Single-step PCR, using the EHC-U and EHC-L primer sets, enabled us to detect *H. pylori* DNA when the target species was mixed 1:1000 with *S. mutans*, a concentration that corresponded to 1 × 10^4^ colony forming units (CFUs)/mL. The nested PCR, using the ET5U and ET-5L primer sets, was 1000 times more sensitive than the single-step PCR, with a detection limit of the target species mixed 1 to 1,000,000 with *S. mutans*, corresponding to 10 CFU/mL (Figure 2). 

## 3. Discussion

We aimed to investigate whether the prevalence of *H. pylori* genomic matter is proportional to that of gastric *H. pylori*. The nested PCR method used in this study could detect oral *H. pylori* in oral samples with no false-positive results if the concentration of the bacterium exceeded 10 CFU/mL, indicating that the primer sets were highly sensitive and specific. Oral *H. pylori* has a characteristic distribution regardless of sex and age, suggesting that *H. pylori* may be a normal component of the microflora in the adult oral cavity. Oral *H. pylori* DNA was frequently detected in patients with a history of gastric *H. pylori* infection.

Since *H. pylori* is detected in the oral cavity, under the setting of gastric infection, the oral cavity may be a potential reservoir for *H. pylori* [8,9,10]. It is generally accepted that *H. pylori* infection occurs mainly in childhood, being transmitted by one’s mother or grandparents, and the bacteria persist in the stomach unless the patient receives eradication therapy [20,21,22]. Although the prevalence of *H. pylori* infection in Japan fell from 10% in individuals born in 1985 to 3% in those born in 2011 [16], we found that the prevalence of oral *H. pylori* did not differ significantly by generation, ranging from 21.4% to 48.3%. Furthermore, the prevalence of oral *H. pylori* DNA was higher than that of gastric *H. pylori* reported in a meta-analysis [17]. These findings suggest that oral *H. pylori* is likely to exist as part of the normal oral microflora. This conclusion is consistent with that of a report by Song et al. [13], who investigated the prevalence of oral and gastric *H. pylori* by nested PCR using the same primer sets in 42 patients who underwent upper gastrointestinal endoscopy due to dyspeptic complaints. Their results indicated that the prevalence of *H. pylori* in the stomach was 26.2%, whereas *H. pylori* DNA prevalence in the oral cavity was 97%, independent of stomach infection status. They similarly concluded that *H. pylori* may be a part of the normal oral microflora.

Disruption of oral microbiota creates dysbiosis, contributing to dental caries, periodontal disease, and an associated increased risk of various diseases such as diabetes, atherosclerotic vascular diseases, and rheumatoid arthritis [23]. A more recent review has demonstrated that oral dysbiosis seems to be more pronounced in patients with tumors of gastrointestinal tract, in particular esophageal, gastric, pancreatic, and colorectal cancers [24]. Although it is unknown how *H. pylori* works as part of the oral microflora, the potential presence of *H. pylori* in the oral cavity may create dysbiosis and increase the risk of developing systemic disease as well as oral disease.

*H. pylori* DNA was frequently detected in the supragingival biofilm on the anterior teeth. The characteristic distribution of oral *H. pylori* may be due to local oxygen concentration and environmental acidity. *H. pylori* prefers microaerobic conditions and is resistant to acidic environments [25,26]. Since anterior teeth are frequently exposed to oxygen, the local oxygen concentration may create a favorable environment for the bacteria. Supragingival plaque may also be an optimum environment for *H. pylori* because the biofilm maintains acidic conditions [27]. In contrast, Song et al. reported that the prevalence of *H. pylori* in the dental plaque from molars, premolars, and incisors was 82%, 64%, and 59%, with ORs of 3.18, 1.24, and 1 (reference), respectively [13]. This discrepancy may be due to the medical history of the participants. Song et al. enrolled patients with dyspeptic complaints, whereas we enrolled healthy participants. Our findings suggest that chronic gastritis may influence local distribution in the oral cavity.

There are several studies showing the strong association of oral hygiene status and oral diseases such as dental caries and periodontitis with oral and gastric *H. pylori* infection [8,10]. Liu et al. reported the relationship between the existence of oral *H. pylori* and the occurrence of dental caries in the oral hygiene index with the same nested PCR method used in this study [28]. Oral *H. pylori* was detected in 126 out of 240 children, and 70 children with dental caries had *H. pylori* in their dental plaque, indicating a statistical correlation between *H. pylori* infection and dental caries or dental hygiene. Dental caries may be an ideal environment for *H. pylori* within the oral cavity. In this study, PI and PD at the sampling site for supragingival dental biofilm in addition to plaque control record score were evaluated. The results showed that oral hygiene status did not link with the prevalence of oral *H. pylori*. This discrepancy may be due to the relative controlled oral hygiene, revealing the mean plaque control record score of 26.7% for all participants.

The amplified *H. pylori* PCR product was detected in 2 of the 88 participants using single-step PCR, and in 32 of the 88 participants following nested PCR, with detection limits of 1 × 10^4^ CFU/mL for single-step PCR and 10 CFU/mL for nested PCR. Although oral *H. pylori* seemed to reside in the oral cavity, the relative frequency was extremely low. Saliva contains bacteria in the order of 10^9^ CFU/mL [29], and 1 g of dental plaque (wet weight) contains 10^11^ CFU/mL [30]. The expected representation of oral *H. pylori* is less than 0.0001% among the normal oral microflora.

Oral *H. pylori* DNA was frequently detected in individuals who had received eradication therapy for gastric *H. pylori*, and the prevalence was statistically higher than that in individuals who did not require eradication therapy. This result indicates that there is an association between gastric *H. pylori* and oral *H. pylori* prevalence. Furthermore, oral *H. pylori* DNA exists in the oral cavity even after eradication therapy. However, the viability of oral *H. pylori* in the commensal oral biofilm remains unclear. We failed to cultivate oral *H. pylori* using the methods described previously [31,32,33,34]. Nevertheless, there is evidence supporting that oral *H. pylori* is viable. One participant who received successful eradication therapy provided supragingival biofilm every two months, and the existence of oral *H. pylori* was examined using the nested PCR. The results showed that oral *H. pylori* DNA was detected three out of six times from either an upper or lower incisor (Appendix A). Since the supragingival biofilm was newly formed on a cleaned tooth surface, oral *H. pylori* may reside somewhere in the oral cavity. In addition, this finding also indicates that oral *H. pylori* is not always detected in carriers. Sampling multiple times is recommended for the accurate detection of oral *H. pylori* DNA.

Although it is demonstrated that oral *H. pylori* DNA is ubiquitous in the oral cavity with or without gastric infection, it is still unclear whether the oral cavity could serve as a potential source of transmission and re-infection. It is also necessary to explore what type of niche is the optimal habitat for *H. pylori*, including subgingival plaque, dental caries and root canal. Further studies are necessary to elucidate the influence of oral *H. pylori* on the oral microbiota, oral disease, and gastric cancer recurrence.

## 4. Materials and Methods 

### 4.1. Participants

Eighty-eight adults (54 women, 34 men; mean age: 52 years, range: 24 to 91 years) were recruited from the patients attending Niigata University Medical and Dental Hospital (Niigata, Japan) between January 2018 and April 2020 for dental check-up or dental disorders. The study protocol was approved by the Niigata University Ethics Committee (approval number 2017-0150), and the methods were carried out in accordance with the approved guidelines. All participants signed an informed consent form before participating in the study. Participants who had at least an incisor and a molar in both the maxilla and mandible were included. None of the volunteers had used systemic antibiotics or antibacterial mouthwashes within three months prior to the start of the study. Past medical history was recorded for each participant via a questionnaire. Of the 88 participants, three were gastric *H. pylori* carriers, who had been diagnosed by routine endoscopic and histological examinations. Twelve participants had received eradication therapy for gastric *H. pylori*. None of all the participants reported upper digestive symptoms. The characteristics of the participants enrolled in this study are summarized in Table 1.

### 4.2. Sampling

Oral hygiene status of participants was estimated according to the O’Leary’s plaque control record before sampling. Unstimulated saliva (2 mL) was collected by having participants spit into a tube. Supragingival dental biofilm samples were scraped from the upper incisors, lower incisors, upper right molars, and lower left molars using a sterile curette. If a participant lacked a tooth in the specified position, the sample was obtained from the opposite side. PI and PD at the sampling site for supragingival dental biofilm were recorded. Each sample was transferred into a tube containing phosphate buffered saline (PBS; pH 7). The superficial layers of the tongue were collected with five gentle strokes from the papillae circumvallatae to the anterior part of the tongue dorsum using a commercially available tongue brush (Tongue Cleaner Plus, Ci Medical, Ishikawa, Japan). The bacteria were retrieved by vigorous stirring in 20 mL of PBS. The samples were centrifuged for 10 min at 10,000 rpm, washed twice with PBS, and stored at −80 °C until DNA extraction. 

### 4.3. DNA Extraction and Nested PCR

DNA was extracted from the saliva, dental plaque, and tongue coat samples, as well as *H. pylori* IID3023 (a gift from Dr. Mimuro, Osaka University, Japan), using the NucleoSpin^®^ Microbial DNA Kit (TaKaRa Bio, Shiga, Japan) according to the manufacturer’s instructions. The quantity and purity of the DNA were assessed by spectrophotometry at 260/280 nm. The DNA was stored at –80 °C until processing.

A nested PCR method established by Song et al., with highly sensitive and specific primer sets and with a low false-positive rate, was used in this study [11,13,35,36]. Briefly, the first round of PCR amplification was performed with Takara Ex Taq^®^ Hot-Start Version (RR006A; TaKaRa Bio, Shiga, Japan) using the EHC-U (5′-CCCTCACGCCATCAGTCCCAAAAA-3′) and EHC-L (5′-AAGAAGTCAAAAACGCCCCAAAAC-3′) primers targeting an 860-bp fragment of *H. pylori* genomic DNA. The amplification comprised 40 cycles of denaturation at 98 °C for 10 s, annealing at 57 °C for 30 s, and extension at 72 °C for 1 min on a MiniAmp™ Thermal Cycler (Thermo Fisher Scientific, Waltham, MA, USA). The expected product size was 417 bp, covering the area from 80,076 bp to 80,492 bp in the *H. pylori* genome [37]. 

For the nested PCR assay, the amplification product (1 µL) obtained by single-step PCR was re-amplified over 20 cycles under the same conditions as in the first round. The internal primer pair ET5U (5′-GGCAAATCATAAGTCCGCAGAA-3′) and ET-5L (5′-TGAGACTTTCCTAGAAGCGGTGTT-3′) was used at a concentration of 50 pmol/µL [36]. The expected size was 228 bp. *H. pylori* IID3023 DNA served as the positive control, and water was used as the negative control. Each PCR product was confirmed by 1.5% agarose gel electrophoresis.

### 4.4. Sequencing of Nested PCR Products

The oral and *H. pylori* IID3023 PCR products were electrophoresed on 1.5% agarose gels containing SYBR^®^ Safe DNA Gel Stain (Thermo Fisher Scientific). The 228-bp fragment was excised from the gel using the FastGene^®^ Agarose Gel Cutter (NIPPON Genetics, Tokyo, Japan), and purified using Freeze ‘N Squeeze™ DNA Extraction Spin Columns (Bio-Rad Laboratories, Hercules, CA, USA). Next-generation sequencing libraries were prepared by fragmenting genomic DNA (16.7 to 17.0 ng) and by ligation with a FastGene Adapter Kit (FG-NGSAD24; Nippon Genetics, Tokyo, Japan) according to the manufacturer’s instructions. The libraries were sequenced by Bioengineering Laboratory (Kanagawa, Japan) using 2 × 300 bp paired-end sequencing on the MiSeq platform (Illumina, San Diego, CA, USA) to analyze DNA homology.

### 4.5. Examination of the Detection Limit of the Single-Step and Nested PCR Methods

To examine the detection limit for oral *H. pylori* using the single-step and nested PCR methods, we mixed the PCR products from the bacterial suspensions of *H. pylori* IID3023 and *Streptococcus mutans* UA159 in various proportions. *S. mutans* and *H. pylori* were suspended in PBS, and the optical density of each suspension was adjusted to 0.5 at 600 nm. We prepared 10-fold serial dilutions of the *H. pylori* bacterial suspension, which we mixed with that of *S. mutans*: the ratio of *H. pylori* to *S. mutans* ranged from 1:1 to 1:10^−7^. DNA was extracted from each mixture, followed by PCR amplification as described previously. The detection limit of *H. pylori* was determined by 1.5% agarose gel electrophoresis.

### 4.6. Statistical Analysis

Data analysis was carried out using SPSS^®^ 11.0 (SPSS, Chicago, IL, USA). The chi-squared test and Fisher’s exact probability test were used when applicable, and the results were considered statistically significant when the *p*-value was <0.05. Prevalence was expressed as a proportion and the crude odds ratio (OR) was used to measure the strength of the association between variables.

A paired Student’s *t*-test was used to compare plaque control record score. The difference in PI between oral *H. pylori* DNA positive and negative individuals was evaluated using the Mann-Whitney *U* test.

## Figures and Tables

**Figure 1 pathogens-10-00010-f001:**
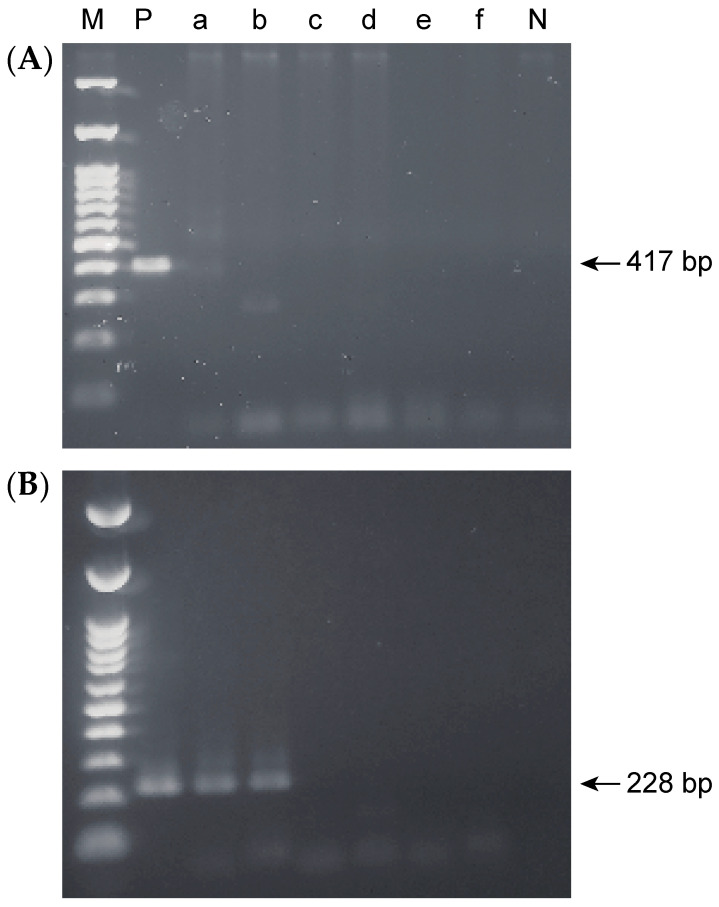
*H. pylori* DNA was detected in oral sampling sites. Agarose gel electrophoresis images. (**A**) Single-step polymerase chain reaction (PCR). (**B**) Nested PCR. Lane M, 100-bp DNA ladder; P, positive control (IID3023); a, upper incisors; b, lower incisors; c, upper right molars; d, lower left molars; e, tongue; f, saliva; N, negative control.

**Figure 2 pathogens-10-00010-f002:**
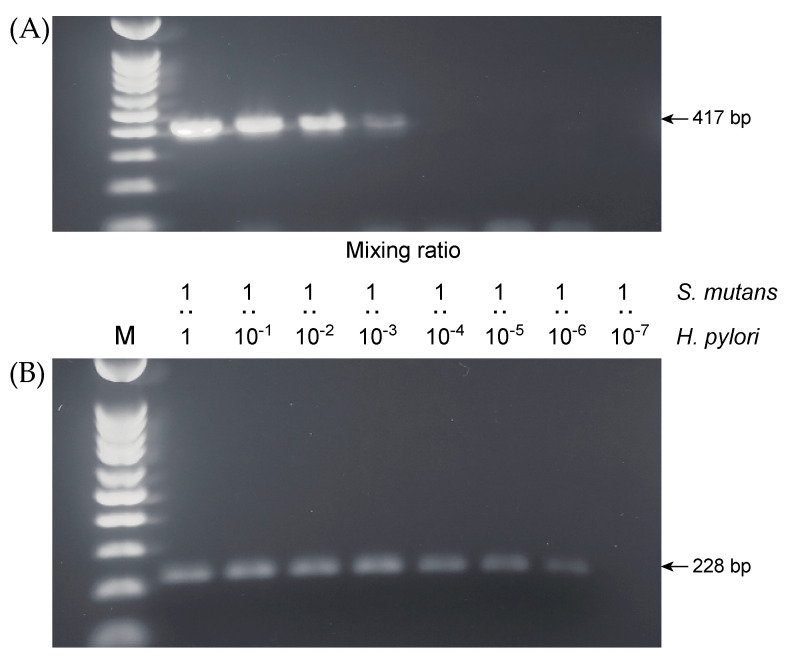
Detection limits for single-step and nested PCR. PCR products were prepared from bacterial suspensions containing various proportions of *H. pylori* IID3023 and *S. mutans* UA159. (**A**) Single-step PCR. (**B**) Nested PCR. Lane M, 100-bp DNA ladder.

**Table 1 pathogens-10-00010-t001:** Characteristics of participants by gender, age and medical history.

Variable	Male	Female	Total
Gender			
N (%)	34 (38.6)	54 (61.4)	88 (100)
Age			
Mean ± SD	46.7 ± 20.8	55.2 ± 20.4	52.1 ± 20.8
Medical history N (%)			
Gastric infection carrier	0 (0)	3 (0.06)	3 (0.03)
Received eradication therapy	3 (0.09)	9 (0.17)	12 (0.14)
Experience not required	31 (91.2)	42 (77.8)	73 (83.0)

**Table 2 pathogens-10-00010-t002:** Association between the presence of oral *H. pylori* and socio-demographic variables.

	Individuals (Positive/Total)	Prevalence (%)	*p*
Gender			
Male	10/34	29.4	
Female	22/54	40.7	0.364
Generation (age)			
Young (24 to 34)	6/28	21.4	
Middle-aged (35 to 64)	14/29	48.3	
Elderly (64 to 91)	12/31	38.7	0.103

**Table 3 pathogens-10-00010-t003:** Association between the presence of oral *H. pylori* and medical history.

	Individuals (Positive/Total)	Prevalence (%)	*p*
Medical history			
Gastric infection carrier	3/3	100	
Received eradication therapy	9/12	75.0	
Experience not required	20/73	27.4	0.02

**Table 4 pathogens-10-00010-t004:** Frequent sites of oral *H. pylori*.

	Individuals (Positive/Total)	Prevalence (%)	OR
Supragingival biofilm			
Lower incisor	19/88	21.6	24.0
Upper incisor	14/88	15.9	16.5
Lower left molar	3/88	3.4	3.1
Upper right molar	1/88	1.1	1
Saliva	4/88	4.5	
Tongue	2/88	2.3	

The number includes overlapping detections among the above individual sites. OR: Odds ratio.

## Data Availability

Data available on request due to restrictions eg privacy or ethical. The data presented in this study are available on request from the corresponding author. The data are not publicly available due to assure participant confidentiality.

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
