# Peer review of "Current Prevalence of Oral Helicobacter pylori among Japanese Adults Determined Using a Nested Polymerase Chain Reaction Assay"

_pathogens, 2020, doi:10.3390/pathogens10010010_

Round 1
Reviewer 1 Report
In the paper entitled “Current prevalence of oral Helicobacter pylori among Japanese adults determined using a nested polymerase chain reaction assay” the authors investigated the presence of Helicobacter pylori in oral cavity of 88 patients by using nested polymerase chain reaction. Results were interesting, showing Helicobacter pylori DNA in the oral cavity of 32 patients. Supragingival biofilm was the most common oral Helicobacter pylori niche. Furthermore, the distribution of this microbe was independent of sex and age, suggesting that it is part of the normal microflora in the adult oral 23 cavity. The techniques utilized were appropriate and described with plenty details. This is a well-designed study with rigorous methods. The discussion is well-balanced and the statements are supported by the data. The study is on a timely subject, confirming the colonization of the oral cavity by Helicobacter pylori. I suggest some minor revision to improve the paper:
- In Results section please report the number of cases and not only the percentages (if n < 100, the use of percentage implies a spurious impression of accuracy).
- Regarding the role of H. pylori in gastric cancer development and the relationship between this pathogen and other bacteria of oral cavity, I suggest considering a recently published review about the relationship between oral microbiota and gastrointestinal tract cancer (for your convenience: doi: 10.3389/fcimb.2019.00232).
- Minor language corrections should be necessary.
Reviewer 2 Report
Major criticism
These researchers determined the presence of H. pylori genomic matter in oral cavity of adult individuals that did not refer symptoms or show signs referable to disorders of the gastroduodenal tract. They reported results according to several variables, not included the occurrence of ailments of the oral cavity, such as caries, paradontopathy etc., or not. Since such pathogen has been associated with the above-mentioned pathologies, I believe it would be important walking this way.
Minor criticisms
Line 13. The authors affirm that they obtained samples from patients without gastric symptoms. Some of them were suffering from dental disorders (patients) and some were attending dentist’s office for dental check-up (subjects) So, I would say “Samples were obtained from 88 subjects and patients attending dentist’s office for dental check-up or dental disorders…” None of them complained upper digestive symptoms”
Line 32. “entero-oral” sounds better than “faecal-oral”, which literally suggests that people eat faecal matter.
Line 39. Correct “0%” in “0”
Line 42. Include citation #11 in square brackets.
Lines 102 and 104. Change “set” with “sets”
Line 102. The following sentence is confused: “Single-step PCR, using the EHC-U and EHC-L primer set, enabled H. pylori DNA to detection when the target species was mixed 1 to 1,000 with S. mutans, a concentration that corresponded to 1 × 104 colony forming units (CFUs)/mL.”
Did the author mean “Single-step PCR, using the EHC-U and EHC-L primer sets, enabled us to detect H. pylori DNA to detection when the target species was mixed 1 to 1,000 with S. mutans, a concentration that corresponded to 1 × 104 colony forming units (CFUs)/mL?
Line 106. Change “1 x 101 in “10”
Line 112. Instead of saying “the prevalence of H. pylori infection …”, I would say “the presence of H. pylori genomic matter”.
Line 121. Instead of saying “through one’s mother or grandparents”, I would say “being transmitted by one’s mother or grandparents”.
Table 3. The expression “Gastric infection career” is not clear. Perhaps the authors meant “Gastric infection carrier”
